# Regulation of Skeletal Development and Maintenance by Runx2 and Sp7

**DOI:** 10.3390/ijms251810102

**Published:** 2024-09-20

**Authors:** Toshihisa Komori

**Affiliations:** Department of Molecular Tumor Biology, Nagasaki University Graduate School of Biomedical Sciences, 1-7-1 Sakamoto, Nagasaki 852-8588, Japan; komorit@nagasaki-u.ac.jp

**Keywords:** Runx2, Sp7, Cbfb, osteoblast differentiation, chondrocyte differentiation, osteoblast proliferation, bone matrix protein genes, osteocalcin, osteocyte processes, osteoarthritis

## Abstract

Runx2 (runt related transcription factor 2) and Sp7 (Sp7 transcription factor 7) are crucial transcription factors for bone development. The cotranscription factor Cbfb (core binding factor beta), which enhances the DNA-binding capacity of Runx2 and stabilizes the Runx2 protein, is necessary for bone development. Runx2 is essential for chondrocyte maturation, and Sp7 is partly involved. Runx2 induces the commitment of multipotent mesenchymal cells to osteoblast lineage cells and enhances the proliferation of osteoprogenitors. Reciprocal regulation between Runx2 and the Hedgehog, fibroblast growth factor (Fgf), Wnt, and parathyroid hormone-like hormone (Pthlh) signaling pathways and Dlx5 (distal-less homeobox 5) plays an important role in these processes. The induction of Fgfr2 (Fgf receptor 2) and Fgfr3 expression by Runx2 is important for the proliferation of osteoblast lineage cells. Runx2 induces Sp7 expression, and Runx2^+^ osteoprogenitors become Runx2^+^Sp7^+^ preosteoblasts. Sp7 induces the differentiation of preosteoblasts into osteoblasts without enhancing their proliferation. In osteoblasts, Runx2 is required for bone formation by inducing the expression of major bone matrix protein genes, including Col1a1 (collagen type I alpha 1), Col1a2, Spp1 (secreted phosphoprotein 1), Ibsp (integrin binding sialoprotein), and Bglap (bone gamma carboxyglutamate protein)/Bglap2. Bglap/Bglap2 (osteocalcin) regulates the alignment of apatite crystals parallel to collagen fibrils but does not function as a hormone that regulates glucose metabolism, testosterone synthesis, and muscle mass. Sp7 is also involved in Co1a1 expression and regulates osteoblast/osteocyte process formation, which is necessary for the survival of osteocytes and the prevention of cortical porosity. SP7 mutations cause osteogenesis imperfecta in rare cases. Runx2 is an important pathogenic factor, while Runx1, Runx3, and Cbfb are protective factors in osteoarthritis development.

## 1. Introduction

Runx2 and Sp7 are transcription factors that play crucial roles in osteoblast differentiation and bone formation, because the germline deletion of Runx2 or Sp7 results in a complete lack of intramembranous and endochondral bone [1,2,3]. Runx2 belongs to the Runx family of transcription factors (Runx1-3), which contain the DNA-binding domain runt [4]. Runx1 and Runx3 are also involved in bone development [5,6,7,8]. Runx family transcription factors are stabilized by and enhance their DNA-binding capacity through the formation of heterodimers with the cotranscription factor Cbfb, which is ubiquitously expressed [4]. Runx2 is expressed in all osteoblast-lineage cells, including osteoprogenitors, preosteoblasts, immature osteoblasts, mature osteoblasts, and osteocytes, as well as in the prehypertrophic, hypertrophic, and terminal hypertrophic chondrocytes. Runx2 is critical not only for osteoblast differentiation but also for chondrocyte maturation [4]. Runx2 expression in osteoblasts and chondrocytes is regulated by enhancers. One of the osteoblast-specific enhancers was activated by the direct binding of Dlx5/6 and Mef2c (monocyte enhancer factor 2C) and indirect binding of Tcf7 (transcription factor 7, T cell specific)/Ctnnb1 (catenin beta 1, β-catenin), Smad1 (SMAD family member 1), Sp7, and Sox5 (sex-determining region Y-box5)/Smad6, forming an enhanceosome [9].

Sp7 belongs to the SP family of transcription factors. Although SP family transcription factors bind GC-rich target sequences through the zinc finger domain, Sp7 appears to indirectly bind to DNA by interacting with homeobox-containing distal-less homeobox (Dlx) factors, such as Dlx5 and Dlx6, which bind AT-rich motifs [9,10]. Runx2 has been shown to induce the expression of Sp7 [11,12]. Runx2 directly regulates Sp7 expression by binding to an enhancer located upstream of Sp7; this enhancer region is also activated by Smad1 [13,14]. Sp7 is expressed in osteoblast-lineage cells, including preosteoblasts, immature osteoblasts, mature osteoblasts, and osteocytes, and in prehypertrophic and terminal hypertrophic chondrocytes [3,11,15,16,17]. Sp7 also plays an important role in chondrocyte maturation [11]. 

This review focuses on the regulation of skeletal development and maintenance by Runx2 and Sp7, including chondrocyte differentiation, osteoblast differentiation, osteoblast proliferation, mature osteoblast functions, diseases caused by RUNX2 and SP7, and the critical roles of Runx family transcription factors in osteoarthritis (OA). 

## 2. Regulation of Chondrocyte Maturation by Runx2 and Sp7

Although Runx2^−/−^ and Sp7^−/−^ mice both completely lack osteoblasts, chondrocyte maturation is inhibited in both mice [1,3,11,18,19]. Previous studies have demonstrated that chondrocyte maturation is severely impaired in Runx2^−/−^ mice, and Col10a1^+^ hypertrophic chondrocytes are absent in most of the skeleton, except for the tibia, fibula, radius, and ulna, where chondrocytes differentiate into terminal hypertrophic chondrocytes with mineralization in the matrix [18,19]. Runx2 was identified as a positive regulator of chondrocyte maturation (Figure 1). Runx3, which is highly expressed in prehypertrophic chondrocytes in the growth plate, is partly involved in chondrocyte maturation, and Runx2^−/−^Runx3^−/−^ mice have been shown to completely lack Col10^+^ mature chondrocytes in their entire skeleton [20,21,22,23]. The requirement of Runx2 for chondrocyte maturation was demonstrated in mice with its deletion in chondrocytes using Col2a1 Cre [24,25]. Runx2 also induced chondrocyte proliferation by upregulating the expression of Ihh [20] (Figure 1). Pthlh, which is induced by Ihh, inhibits chondrocyte maturation through Pth1r in prehypertrophic chondrocytes [26]. Pthlh also downregulates the expression of Runx2 through Pth1r in prehypertrophic chondrocytes, forming a negative feedback loop for chondrocyte maturation [4,27,28,29]. Furthermore, Runx2 has been shown to regulate the expression of cell cycle-related genes, including Gpr132 (G protein-coupled receptor 132), Sfn (stratifin), Myb (myeloblastosis oncogene), and Ccna1 (cyclin A1), in chondrocytes [25].

Previous studies reported that vascular invasion into cartilage, including calcified cartilage, was completely absent in Runx2^−/−^ mice, and the expression levels of Vegfa (vascular endothelial growth factor A) were decreased in the terminal hypertrophic chondrocytes of Runx2^−/−^ mice [30,31]. Since vascular invasion into the cartilage of mice with Vegfa deletion in chondrocytes and perichondrial cells using Col2a1 Cre was delayed [32], the downregulated expression of Vegfa in terminal hypertrophic chondrocytes in Runx2^−/−^ mice was considered to be responsible for the lack of vascular invasion [30,31]. However, vascular invasion into the cartilage occurred in mice with Runx2 deletion in hypertrophic chondrocytes using Col10a1 Cre (Runx2^fl/flCol10a1-Cre^), even though the expression of Vegfa was downregulated in terminal hypertrophic chondrocytes [15]. Vegfa is highly expressed in terminal hypertrophic chondrocytes and osteoblasts. Osteoblast differentiation occurs in the perichondrium of Runx2^fl/flCol10a1-Cre^ mice, and Vegfa is highly expressed in perichondrial osteoblasts [15]. Therefore, Runx2 regulates the expression of Vegfa in terminal hypertrophic chondrocytes. However, the expression of Vegfa in osteoblasts in the perichondrium (bone collar) is sufficient for vascular invasion into the cartilage. In addition to Vegfa, Runx2 has been shown to regulate the expression of Spp1, Ibsp, and Mmp13 (matrix metallopeptidase 13) in terminal hypertrophic chondrocytes, because their expression was downregulated in the terminal hypertrophic chondrocytes of Runx2^fl/flCol10a1-Cre^ mice [15] (Figure 1). 

The transdifferentiation of terminal hypertrophic chondrocytes into osteoblasts or stromal cells has been demonstrated using Col10a1 Cre [33,34]. This transdifferentiation was absent in Runx2^fl/flCol10a1-Cre^ mice, demonstrating the importance of Runx2 in this process [15] (Figure 1). Since the bone volumes of the femur and vertebrae were normal in Runx2^fl/flCol10a1-Cre^ mice after the age of six weeks, transdifferentiation is physiologically important during the embryonic and neonatal stages [15]. 

Chondrocyte maturation was also retarded in Sp7^−/−^ mice, but to a lesser extent than in Runx2^−/−^ mice, as shown by the presence of mineralized cartilage in most endochondral bones, which indicates the maturation of chondrocytes to terminal hypertrophic chondrocytes [3]. The retardation of chondrocyte maturation was also demonstrated in Sp7 conditional knockout mice using Prrx1 (paired related homeobox 1) Cre, which is expressed in mesenchymal cells in the calvaria and limbs, Col2a1 Cre, and Col11a2 Cre, which is expressed in chondrocytes [11] (Figure 1).

## 3. Regulation of Osteoblast Differentiation by Runx2 and Sp7 

Although Runx2^−/−^ and Sp7^−/−^ mice lack osteoblasts, Runx2 is expressed in mesenchymal cells, which are in the presumptive bone region in Sp7^−/−^ mice [3,35]. Runx2 is the first transcription factor that directs multipotent mesenchymal cells to osteoblast-lineage cells; however, the expression of Ihh in prehypertrophic chondrocytes is required for Runx2 expression in perichondrial mesenchymal cells during endochondral ossification [36] (Figure 1). Runx2 is known to induce the expression of Sp7, and Runx2^+^ osteoprogenitors become Runx2^+^Sp7^+^ preosteoblasts [4]. Canonical Wnt signaling is necessary for the differentiation of Runx2^+^Sp7^+^ preosteoblasts into immature osteoblasts, which express Col1a1 and Spp1 [37,38,39,40]. In the differentiation of suture mesenchymal cells into osteoblasts, Runx2 induces their differentiation into osteoblasts through the induction of the expression of Hedgehog, Fgf, Wnt, and Pthlh signaling pathway genes and Dlx5 [41]. Furthermore, these signaling pathways and Dlx5 regulate the expression and/or functions of Runx2 [4]. Therefore, reciprocal regulation plays an important role in the differentiation of mesenchymal cells into osteoblasts. Runx2 and Sp7 are both required for the further differentiation of immature osteoblasts to mature osteoblasts, in which Bglap/Bglap2 and high levels of Col1a1 are expressed [17,42,43] (Figure 1). 

## 4. Regulation of the Proliferation of Osteoprogenitors by Runx2

In Runx2^−/−^ mice, few mesenchymal cells are present in the presumptive bone regions and do not actively proliferate. In contrast, mesenchymal cells are abundant in the presumptive bone regions of Sp7^−/−^ mice, express Runx2, and actively proliferate [35]. The number of actively proliferating osteoprogenitors was shown to be increased in Sp7^−/−^ zebrafish [44]. Furthermore, the number and proliferation rate of mesenchymal cells in Sp7^−/−^ mice were dependent on the gene dosage of Runx2 because the number and proliferation rate of mesenchymal cells in Sp7^−/−^ Runx2^+/−^ mice were approximately 50% of those in Sp7^−/−^ mice. Runx2 induces the expression of Fgfr2 and Fgfr3 and enhances the proliferation of osteoblast progenitors [35] (Figure 1). The Hedgehog, Wnt, and Pthlh signaling pathways are also involved in the proliferation of osteoblast progenitors [41]. 

## 5. Functions of Runx2 and Sp7 in Immature and Mature Osteoblasts

### 5.1. Deletion of Runx2 in Immature Osteoblasts Using 2.3 kb Col1a1 Cre (Runx2^fl/fl;Col1a1-Cre^ Mice) or 2.3 kb Col1a1 GFP-Cre (Runx2^fl/fl;Col1a1-GFP-Cre^ Mice) 

Takarada et al. generated Runx2^fl/fl;Col1a1-Cre^ mice by deleting the DNA-binding domain runt using a 2.3 kb Col1a1 Cre line; however, these mice showed no phenotype [24]. Adhami et al. generated Runx2^fl/flCol1a1-Cre^ mice by deleting exon 8 in the same Cre line used by Takarada. The deletion of exon 8 resulted in a cryptic Runx2 protein that maintained its DNA-binding capacity but had a lower capacity for transcriptional activation, and trabecular and cortical bone volumes were reduced [45]. The crypted Runx2 protein appears to have inhibited the binding of Runx1 and Runx3 to Runx-binding sites, leading to interruption of their functions, because Runx1 and Runx3 are involved in bone development [5,7,8].

Our group generated 2.3 kb Col1a1 green fluorescent protein (GFP)-Cre mice and Runx2^fl/fl;Col1a1-GFP-Cre^ mice, in which the runt domain was deleted in immature osteoblasts [46]. Runx2^fl/fl;Col1a1-GFP-Cre^ mice had a smaller body size, shorter clavicles, and larger unmineralized area in the calvaria, indicating the involvement of Runx2 in skeletal development by functioning not only in an early stage but also in a late stage of osteoblast differentiation [46] (Figure 1). The incisors of Runx2^fl/flCol1a1-GFP-Cre^ mice, 37% of which had abnormal alignment, were shorter than those of control mice. Therefore, Runx2 is also involved in tooth development in both late and early stages. Furthermore, trabecular bone in the femur and vertebrae and femoral cortical bone were decreased in Runx2^fl/fl;Col1a1-GFP-Cre^ mice owing to reduced bone formation. As the number of osteoblasts and their proliferation were reduced in Runx2^fl/fl;Col1a1-GFP-Cre^ mice, Runx2 positively regulated the proliferation of immature osteoblasts (Figure 1). Moreover, the levels of serum markers for bone resorption, tartrate-resistant acid phosphatase 5b (TRAP5b), and C-terminal cross-linked telopeptide of type 1 collagen (CTX1) were reduced, which may have been due to a decrease in the ratio of Tnfsf11 (tumor necrosis factor superfamily, member 11)/Tnfrsf11b (tumor necrosis factor receptor superfamily, member 11b). Osteoblasts were less cuboidal than those in the control mice, and the expression of Col1a1, Col1a2, Spp1, Ibsp, and Bglap/Bglap2 was downregulated. Col1a1 expression was regulated by Runx2, at least partly, through the 2.3 kb Col1a1 promoter, because GFP-Cre expression in Runx2^fl/fl;Col1a1GFP-Cre^ mice was reduced [46]. Therefore, Runx2 induces the proliferation and maturation of immature osteoblasts and regulates the expression of major bone matrix protein genes in immature and mature osteoblasts (Figure 2).

One Runx2 target gene, Galnt3 (UDP-N-acetylgalactosamine (GalNAc): polypeptide GalNAc-transferase 3), which is expressed in osteoblasts and osteocytes in the bone [47], inhibits the proteolytic processing of Fgf23 by O-linked glycosylation. Fgf23, which is expressed in osteocytes, is a hormone that reduces serum phosphorus levels by inhibiting the reabsorption of phosphate in the kidneys and renal biosynthesis of 1, 25(OH)_2_D_3_ [48]. The expression levels of Galnt3 and Fgf23 in Runx2^fl/fl;Col1a1-Cre^ mice were approximately 50% of those in Runx2^fl/fl^ mice, whereas the serum levels of intact Fgf23 and phosphorus were normal in Runx2^fl/fl;Col1a1-Cre^ mice, which may be due to the feedback upregulation of the Fgf23 protein at the translational level. Runx2 is expressed not only in osteoblasts but also in osteocytes. Runx2 regulates the expression of Galnt3 and Fgf23 in osteocytes and is involved in phosphorus homeostasis [47] (Figure 2).

### 5.2. Deletion of Sp7 Using 2.3 kb Col1a1-Cre, 2.3 kb Col1a1-CreERT2, 2.3 kb Col1a1-GFP-Cre, CAG-CreER, or Dmp1 (Dentin Matrix Protein 1)-Cre

Baek et al. generated Sp7^fl/−;Col1a1Cre^ mice using heterozygous germline-deleted mice, and Sp7^fl/−;Col1a1-Cre^ mice were compared with Sp7^fl/+;Col1a1-Cre^ mice. The trabecular bone in the vertebrae of Sp7^fl/−;Col1a1-Cre^ mice was reduced due to a decrease in bone formation [42]. In femurs, trabecular bone volume increased with higher trabecular numbers, but reduced trabecular thickness, and BrdU-positive cell numbers were elevated. Bone resorption in Sp7^fl/−;Col1a1-Cre^ mice was similar to that in Sp7^fl/+;Col1a1-Cre^ mice as shown by TRAP staining and urinary deoxypyridinoline. The expression of Alpl (alkaline phosphatase, liver/bone/kidney) and Spp1 increased in Sp7^fl/−;Col1a1-Cre^ mice, whereas that of Bglap/Bglap2 decreased and that of Col1a1 remained unchanged. The same group also generated Sp7^fl/−Col1a1-CreERT2^ mice, in which 2.3 kb Col1a1 CreERT2 was induced by injecting 4-hydroxytamoxifen after birth [43]. The trabecular bone in the vertebrae decreased because of the reduced bone formation. The number of osteoblasts, osteoclasts, and BrdU-positive cells in the tibiae was similar to that in control mice, and the expression of Ibsp, Col1a1, and Bglap/Bglap2 was lower than that in the control, whereas the expression of Alpl was similar. Sp7^fl/fl;Col1a1-Cre^ mice generated by Moon et al. showed reduced femur length [49]. The metaphyseal cortical bone is formed through the coalescence of the trabecular bone into the cortical bone [50]. Moon et al. indicated that Sp7 regulates the migration of osteoblasts to peripheral spongiosa by directly inducing Itgb3 (integrin beta 3) expression and regulates longitudinal bone growth by enhancing the coalescence of trabecular bone into cortical bone [49]. 

Sp7 comprises two exons. To generate Sp7^fl/fl^ mice, the neo gene was inserted into the intron (Sp7^flneo/flneo^ mice) [17]. We initially analyzed Sp7^flneo/flneo^ mice. The insertion of neo disturbed splicing and reduced Sp7 mRNA expression to 30% of that in wild-type mice. Body weight was reduced in Sp7^flneo/flneo^ mice of both sexes. The trabecular bone volume in femurs increased in female Sp7^flneo/flneo^ mice but not in males. The femoral cortical bone was porous, and its thickness was reduced in both males and females. After deletion of neo, Sp7^fl/+^ or Sp7^fl/fl^ mice were crossed with 2.3 kb Col1a1 GFP-Cre transgenic mice [17]. Femoral trabecular bone increased in female Sp7^fl/fl;Col1a1-GFP-Cre^ mice but not in males. Cortical porosity and reduced femur thickness were observed in both the male and female mice. However, the vertebral trabecular bone was reduced in Sp7^fl/fl;Col1a1-GFPCre^ mice of both sexes. The number of osteoblasts and the bone formation rate both increased in the trabecular bone of the tibiae but decreased in that of the vertebrae, while osteoclast parameters were unchanged in both tissues. The proliferation of osteoblastic cells was also increased in the femurs. Furthermore, bone formation in the femoral cortical bone of Sp7^fl/fl;Col1a1-GFPCre^ mice was similar to that of Sp7^fl/fl^ mice. Spp1 expression was increased, mature osteoblasts were less cuboidal, and Col1a1 and Bglap/Bglap2 expressions were reduced in Sp7^fl/fl;Col1a1-GFPCre^ mice. As GFP-Cre expression was reduced in Sp7^fl/fl;Col1a1-GFPCre^ mice, Col1a1 expression was also regulated by Sp7, at least partly, through the 2.3 kb promoter [17] (Figure 2). These findings indicate that osteoblast maturation was inhibited, whereas bone formation was promoted in the long bones owing to the accumulation of immature osteoblasts, which compensated for interrupted maturation [17]. Therefore, Sp7 enhances the maturation of immature osteoblasts and inhibits their proliferation (Figure 2). Furthermore, osteoblast proliferation in the long bones of Sp7^fl/fl;Col1a1-GFPCre^ mice differed between males and females, indicating that sex hormones, including estrogen and progesterone, affect the proliferation of immature osteoblasts [51,52,53,54,55]. Osteoblast numbers and bone formation were reduced in the vertebrae of Sp7^fl/flCol1a1-GFP-Cre^ mice, indicating differences in the regulation of osteoblast proliferation between long bones and vertebrae. Vertebral skeletal stem cells are distinct from other skeletal stem cells with different expression profiles [56]. This may explain the different phenotypes of vertebrae and long bones. 

The porous cortical bone in Sp7^fl/fl;Col1a1-GFPCre^ mice was caused by enhanced bone resorption due to osteocyte apoptosis and secondary necrosis [17] (Figure 2). Osteocyte apoptosis is attributed to a reduction in the number of osteocyte processes [57]. The number of osteocyte processes was also reduced in Sp7^fl/−;CAG-CreER^ mice, in which Sp7 was deleted in whole tissues by an injection of tamoxifen after birth, and in Sp7^fl/fl;Dmp1-Cre^ mice, in which Sp7 was deleted in osteocytes and mature osteoblasts [16,58]. A reduction in osteocyte processes was also observed in Sp7-overexpressing mice in osteoblasts using the 2.3 kb Col1a1 promoter [52]. Sp7 appears to regulate the formation of these processes by the transitional stage from osteoblasts to osteocytes because it is unlikely to increase osteocyte processes and canaliculi in mineralized bone. Ostn (osteocrin), a small secreted protein that is expressed in osteoblasts in the periosteum [59,60], has been shown to play a role in reducing the number of osteocyte processes in Sp7^fl/fl;Dmp1-Cre^ mice [58]. However, Ostn expression was not downregulated in the femurs of Sp7^fl/fl;Dmp1-Cre^ mice [58]. Furthermore, Ostn expression levels were increased in the osteoblast-enriched fraction from Sp7^fl/fl;Col1a1-GFPCre^ mice, and their levels in the osteocyte-enriched fraction were similar to those in the control mice [17]. Moreover, Ostn expression levels were similar in osteoblast- and osteocyte-enriched fractions from Sp7-overexpressing and wild-type mice [17]. Therefore, the mechanisms by which the formation of osteocyte processes is regulated warrant further investigation.

Sost (sclerostin) expression in osteocytes was reduced in Sp7^fl/−;CAG-CreER^, Sp7^fl/fl;Col1a1Cre^, and Sp7^fl/fl;Col1a1-GFPCre^ mice [16,17,61], and Sp7 was shown to directly regulate Sost expression [16]. However, the expression of other osteocyte marker genes, including Dmp1, Phex (phosphate-regulating endopeptidase homolog, X-linked), Fgf23, and Mepe (matrix extracellular phosphoglycoprotein with ASARM motif), was also reduced in osteocytes in these mice [16,17,61]. Moreover, the frequency of Sost-positive osteocytes and the serum level of Sost were reduced in Sp7-overexpressing mice using the 2.3 kb Col1a1 promoter. A common feature of these mice is a reduction in the number of osteocyte processes. The lacunocanalicular network is responsible for mechano-sensing and mechano-transduction, and Sost expression changes upon mechanical loading [62,63,64,65,66]. Thus, disruption of the lacunocanalicular network appears to have caused a reduction in Sost expression. Indeed, reduced Sost expression may have contributed to the increase in bone formation and proliferation of periosteal osteoprogenitors in Sp7^fl/fl;Col1a1Cre^ mice and Sp7^fl/fl;Col1a1-GFPCre^ mice [17,61]. 

Data from these Sp7 conditional knockout mice demonstrated that Sp7 inhibits the proliferation of immature osteoblasts, induces their maturation and Col1a1 expression, and regulates the formation of osteoblast/osteocyte processes (Figure 2). 

### 5.3. Functions of Ocn, the Expression of Which Is Regulated by Runx2

Ocn is an important target of Runx2. It is encoded by three genes, Bglap, Bglap2, and Bglap3. Bglap and Bglap2, but not Bglap3, are specifically expressed in mature osteoblasts. Ocn is the most abundant non-collagenous protein in the bone. It exhibits a high affinity for Ca_2_^+^ via the carboxylation of three glutamic acids. Uncarboxylated Ocn has a low affinity for Ca_2_^+^ and enters circulation. Mice with the deletion of Bglap and Bglap2 were generated by Karsenty’s group and exhibited markedly increased trabecular and cortical bone volumes due to enhanced bone formation, indicating that Ocn inhibits bone formation [67]. However, an increase in cortical thickness was not observed in the same Ocn^−^/^−^ mouse line [68]. Furthermore, Ocn^−^/^−^ mice show impaired glucose metabolism, reduced testosterone synthesis, impaired spermatogenesis, reduced muscle mass, profound deficits in special learning and memory, and an exacerbation of anxiety-like behavior [69]. Therefore, uncarboxylated Ocn has been shown to function as a hormone that regulates glucose metabolism, testosterone synthesis, muscle mass, brain development and function, and parasympathetic tone. Our group and William’s group have recently and independently generated Ocn^−^/^−^ mice. Ocn^−^/^−^ mice generated by the two groups showed normal trabecular and cortical bone masses in both sexes at all ages and similar bone formation to that of wild-type mice [70,71]. Furthermore, glucose metabolism, testosterone synthesis, and spermatogenesis were normal in Ocn^−^/^−^ mice in both the groups (Figure 2). Glucose metabolism is not impaired in Ocn^−^/^−^ rats [72]. Moreover, muscle mass was normal in Ocn^−^/^−^ mice [71]. These discrepancies among Ocn^−^/^−^ mouse lines can be attributed to the ignorance of genetic backgrounds, because knockout and control mice with inappropriate genetic backgrounds were used and compared, as shown in experiments on glucose metabolism, in which Ptprv^−^/^−^ (protein tyrosine phosphatase receptor type V) mice (outbred MF1 strain) and osteoblast-specific Ptprv conditional knockout mice (129/Sv:FVB/N strain) were compared with a single control group with an unspecified genetic background [73]. Furthermore, the genetic backgrounds differed from those in the literature and among studies published by the same group without a description of backcrossing [74,75,76,77]. Because glucose metabolism markedly differs among strains [78], their genetic backgrounds need to be considered. 

In wild-type mice, collagen fibers run parallel to the longitudinal direction of the long bones, and apatite crystals are aligned parallel to the collagen fibrils. Although the direction of the collagen fibers was parallel to the longitudinal direction of the long bones in Ocn^−^/^−^ mice, the alignment of apatite crystals was random (Figure 2). Therefore, Ocn is required for the alignment of apatite crystals parallel to collagen fibrils [71]. 

## 6. Cbfb, a Cotranscription Factor for Runx2

Cbfb does not exhibit DNA-binding activity but forms a heterodimer with Runx family transcription factors and plays an important role in the transcriptional activities of Runx family transcription factors by enhancing their DNA-binding activity. Cbfb^−^/^−^ mice died at E11.5–13.5 and lacked definitive hematopoiesis in the fetal liver, and these phenotypes were similar to those in Runx1^−/−^ mice, demonstrating the importance of Cbfb in the function of Runx1 [79,80]. Three different approaches rescued embryonic lethality due to the lack of hematopoiesis; however, these mice died at birth due to severely impaired endochondral and intramembranous ossification caused by marked delays in osteoblast differentiation and chondrocyte maturation, indicating that Cbfb is also required for Runx2 [81,82,83]. 

Cbfb functions were examined by generating conditional knockout mice using Sp7-Cre, 2.3 kb Col1a1-Cre, Prrx1-Cre, Twist2 (twist basic helix–loop–helix transcription factor 2)-Cre, and Col2a1-Cre. Although these mice showed similar phenotypes to Runx2^−/−^ mice or Runx2 conditional knockout mice, the phenotypes of Cbfb conditional knockout mice were milder than those of Runx2 conditional knockout mice or Runx2^−/−^ mice [84,85,86,87,88,89]. Cbfb not only enhances the DNA-binding capacity of Runx2 but also stabilizes Runx2 protein [88,89]. 

Cbfb has two functional isoforms, Cbfb1 and Cbfb2, which are formed by alternative splicing. The expression of Cbfb2 was approximately three-fold higher than that of Cbfb1 and was upregulated in Cbfb1^−/−^ mice, whereas no increase was observed in the expression of Cbfb1 in Cbfb2^−/−^ mice. Osteoblast differentiation and chondrocyte maturation were inhibited in Cbfb2^−/−^ mice but not in Cbfb1^−/−^ mice; however, Cbfb1 more strongly enhanced Runx2 activity [90]. Therefore, the relative levels of these isoforms appear to regulate skeletal development with functional redundancy by adjusting the capacity of Runx2 for transcriptional activation to the appropriate physiological levels. Furthermore, a comparison of the phenotypes of Cbfb2^−/−^ mice and Runx2^+/−^ mice indicated that different amounts of Cbfb and Runx2 were required among the calvaria, limbs, vertebrae, ribs, and clavicles for bone development and osteoblast proliferation and differentiation. Moreover, the dependency of Runx2 protein stability on Cbfb differs among skeletal tissues [91]. 

## 7. Pathogenesis of Human Skeletal Diseases with RUNX2 or SP7 Mutations

### 7.1. Cleidocranial Dysplasia (CCD) Caused by Heterozygous Mutations of RUNX2

Heterozygous mutations in RUNX2 cause cleidocranial dysplasia (CCD), which is characterized by hypoplastic clavicles, open fontanelles, supernumerary teeth, and short stature [92]. Runx2^+/−^ mice are similar to human CCD, except for the teeth, because mice have only primary teeth. The lengths of the long bones and vertebrae in Runx2^+/−^ mice are shorter, and the development of the calvaria and clavicles is severely impaired. Although bone formation is reduced in both the femoral and vertebral trabecular bones, a reduction in trabecular bone volume is apparent in the femurs but not in the vertebrae. The femoral cortical thickness is also reduced, and the bone mineral density is reduced in the trabecular bone in both femurs and vertebrae and in the femoral cortical bone. Bone resorption is also reduced in Runx2^+/−^ mice because Runx2 regulates Tnfsf11 expression [93]. Thus, whole bones are affected by haploinsufficiency of Runx2. 

However, the reason why the development of the calvaria and clavicles is severely impaired in CCD has been a big puzzle for a long time. Runx2 directly regulated the expression of Hedgehog, Fgf, Wnt, and Pthlh signaling pathway genes, including Gli1 (GLI-Kruppel family member GLI1), Ptch1 (patched 1), Ihh, Fgfr2, Fgfr3, Tcf7, Wnt10b, and Pth1r, and a transcription factor Dlx5 in osteoblasts. More than half the dosage of Runx2 was necessary for their induction in suture mesenchymal cells, whereas half the dosage of Runx2 was sufficient for their induction in committed osteoblasts and Runx2^+/−^ osteoblasts normally expressed bone matrix protein genes. Moreover, the activation of these signaling pathways is required for the proliferation and commitment of suture mesenchymal cells to osteoblast-lineage cells. Thus, Runx2 regulates calvarial development by inducing these signaling pathway genes and Dlx5, and more than half of the dosage of Runx2 is necessary for the proliferation and commitment of suture mesenchymal cells into osteoblast-lineage cells [41,91]. The Fgf signaling pathway activates and stabilizes Runx2 protein through phosphorylation by Mapk1/3 (mitogen-activated protein kinase 1/3), Tcf7/Ctnnb1, Dlx5, and Sp7 activate the promoter and osteoblast-specific enhancer of Runx2, and Pth (parathyroid hormone) increases Runx2 expression and activity [9,94,95,96,97,98,99]. Therefore, reciprocal regulation between Runx2 and these signaling pathways and transcription factors is also important for calvarial development.

### 7.2. Osteogenesis Imperfecta (OI) Caused by SP7 Mutations

OI is a heterogeneous group of inherited connective tissue disorders characterized by bone fragility and deformity and other anomalies, including short stature, opalescent teeth, blue sclerae, and hearing loss [100]. The majority of patients with OI have autosomal dominant variants of COL1A1 and COL1A2. Four mutations in SP7 have been reported to cause recessive or dominant OI (OI type XII) [101,102,103,104,105,106]. Patients homozygous for the 946C>T (R316C) mutation of SP7 show high bone turnover and cortical porosity, which is similar to the phenotypes of Sp7^fl/fl;*Col1a1*-GFPCre^ mice [17,102]. The increased bone formation in the R316C mutant appears to be caused by accumulated immature osteoblasts and reductions in SOST expression and its secretion to the bone surface, as seen in Sp7^fl/fl;*Col1a1*-GFPCre^ mice. As the length and number of osteocyte processes are reduced in R316C patients, cortical porosity is likely to be caused by osteocyte apoptosis, as seen in Sp7^fl/fl;Dmp1-Cre^ mice and Sp7^fl/fl;*Col1a1*-GFPCre^ mice [17,58]. Sibling patients with heterozygous mutations of 1019A>C (E340A) showed low turnover and cortical porosity. As the patients suffered frequent and severe fractures irrespective of the increased or normal level of trabecular volumetric bone mineral density in the radii, the mutation is likely to cause a more severe impairment of osteoblast differentiation than in Sp7^fl/fl;*Col1a1*-GFPCre^ mice. Thus, the E340A mutation in SP7 appears to function as a dominant negative form of SP7. Although bone biopsy was not performed in E340A patients, cortical porosity is likely to be caused by a reduction in osteocyte processes. Patients with the other mutations in SP7, 1052delA (E351GfsX19) and 824G>A (C257Y), showed features of recessive OI, but the bone phenotypes have not been reported in detail [101,103,107]. 

In contrast, patients with the heterozygous 926C>G (Ser309Trp) mutation in SP7 showed osteosclerosis, hyperostosis, and long bone fragility with high bone turnover [108,109]. This mutation alters the DNA binding specificity of SP7 from AT-rich motifs to the GC-consensus sequence, resulting in aberrant target gene activation [109]. Moreover, a patient with two heterozygous frameshift deletions in SP7 showed combined features of OI and sclerotic skeletal dysplasia [110]. 

## 8. Progression or Protection of OA by Runx Family Transcription Factors and Cbfb

### 8.1. Runx2 Is a Major Pathogenic Factor for OA 

OA is a degenerative disease of articular cartilage characterized by synovial hyperplasia, articular cartilage degradation, subchondral sclerosis, and osteophyte formation [111]. In OA, Prg4 (proteoglycan 4, lubricin), which is a lubricant of joints and maintains superficial chondrocytes, and cartilage matrix proteins such as type II collagen and Acan (aggrecan) are reduced, and matrix-degrading enzymes are released from chondrocytes (Figure 3). Chondrocyte maturation induces these processes. Matrix metalloproteinases (MMPs, especially MMP13) and a disintegrin and metalloproteinases with thrombospondin motifs (ADAMTSs), especially ADAMTS5, facilitate type II collagen and aggrecan degradation, respectively [112]. Runx2 is an essential transcription factor for chondrocyte maturation and induces the expression of Mmp13 and Adamts5 [4,99,113,114,115,116,117,118]. Interactions between Runx2 and AP1 and between Runx2 and Cebpb (CCAAT/enhancer binding protein beta) for Mmp13 induction have been reported [116,118]. Furthermore, chondrocyte-specific Smad3 deletion using Col2a1 Cre resulted in progressive cartilage degradation, and Smad3 decreased Runx2-induced Mmp13 expression, indicating that Tgfβ signaling through Smad3 maintains the balance between cartilage matrix synthesis and degradation [119]. Furthermore, single nucleotide polymorphisms (SNPs) in the RUNX2 locus are associated with OA [120,121]. 

Runx2^+/−^ mice exhibited decreased articular cartilage destruction in a mouse surgical OA model [123]. Furthermore, articular cartilage destruction in a surgical OA model was ameliorated in tamoxifen-induced Runx2-deleted mice in chondrocytes using Acan-CreER [124]. Moreover, chondrocyte-specific Runx2 overexpression accelerates the progression of articular cartilage degradation in a mouse surgical OA model [125]. However, in tamoxifen-induced Runx2 deletion in chondrocytes using Col2a1-CreERT2, OA development was inhibited in heterozygously deleted mice but accelerated in homozygously deleted mice [126]. Runx2 may be involved in Col2a1 transcription through the activation of Col2a1 enhancers and the promoter [126]. Thus, the level of Runx2 suppression required for the inhibition of OA development requires further investigation. 

The transcriptional coactivator Yap1, a key molecule in the Hippo signaling pathway, controls cell proliferation and differentiation [127]. Yap1 inhibits chondrocyte maturation by inhibiting Runx2 through protein–protein interactions [128]. The adaptor protein Shc1 (src homology 2 domain-containing transforming protein C1) binds to the cytoplasmic tail of growth factor receptors or the receptors for extracellular matrix when they are activated, and it activates the Ras (rat sarcoma virus oncogene)–Raf (v-raf-leukemia viral oncogene)–Map2k1 (mitogen-activated protein kinase kinase 1)–Mapk1/3 pathway or connects the extracellular matrix and cytoskeleton [129]. Mice with Shc1 deletion using Twist2 Cre, which deletes target genes in osteochondroprogenitor cells, were protected from age-associated OA development. Shc1 promotes Mapk1/3 activation and the nuclear translocation of Runx2 and maintains Yap1 in its inactive form [130]. Fermt2 is an essential focal adhesion protein. Tamoxifen-induced Fermt2-deletion in chondrocytes using Acan-CreERT2 caused spontaneous OA and exacerbated surgical OA, and the heterozygous deletion of Runx2 in chondrocytes reversed OA development by the surgical OA of Fermt2-deleted mice in chondrocytes. Fermt2 deficiency upregulates Runx2 expression through Stat3 activation [131]. 

### 8.2. Runx1 Protects OA Development

Runx1 is expressed in the resting, proliferating, and prehypertrophic layers of cartilage during skeletal development [132]. In articular cartilage, Runx1 is enriched in the superficial zone of articular cartilage [133]. Yano et al. showed that Runx1 expression is reduced in the cartilage in a surgical OA model mice, and Runx1 directly regulates Col2a1 expression and suppresses Col10a1 expression. Furthermore, a thienoindazole derivative compound, TD-198946, induced Runx1 and Col2a1 but not Col10a1 in vitro, and its intra-articular injection in a surgical OA mouse model increased Runx1, Sox9, and Sox6 expression but reduced Col10a1 expression in articular chondrocytes, and it also prevented and repaired OA. Thus, TD-198946 is thought to exert its effects through Runx1 expression [134]. Furthermore, the intra-articular injection of polyplex nanomicelles carrying RUNX1 mRNA prevents the progression of surgical OA [135,136]. The chondrocyte-specific deletion of Runx1 using Col2a1-Cre accelerated surgical OA, and Runx1 enhanced cartilage matrix production by directly interacting with Sox5, Sox6, and Sox9, and suppressed chondrocyte maturation through the induction of Nkx3-2 [137]. Zhou et al. also showed that the chondrocyte-specific deletion of Runx1 using Col2a1 Cre aggravates surgical OA. Furthermore, it affected the growth plate. Col2a1 and Sox9 expression was reduced, chondrocyte proliferation was reduced, Mmp13 expression was increased, and subchondral bone was reduced in Runx1^fl/fl;Col2a1-Cre^ mice. The intra-articular injection of Runx1-expressing AAV (adeno-associated virus) alleviated OA progression and the abnormalities in the growth plate and subchondral bone. Tapt1 (transmembrane anterior posterior transformation 1), Ric1 (RAB6A GEF complex partner 1), and Fgf20 were identified as Runx1 targets in the articular cartilage and the growth plate [138]. Zhang et al. also showed that the chondrocyte-specific deletion of Runx1 using Col2a1-Cre or Col2a1-CreER causes spontaneous OA and aggravates surgical OA. The loss of Runx1 in cartilage decreased YAP and p-Smad2/3 levels and increased active β-catenin levels. Runx1 appears to attenuate OA progression by regulating the Hippo/Yap, Wnt/β-catenin, and Tgfβ/Smad2/3 signaling pathways. AAV-Runx1 treatment also protected against articular cartilage damage in surgical OA [139]. In contrast to a previous study [138], a reduction in Tapt1 and Ric1 expression was not observed in the hip articular cartilage of Runx1^fl/fl;Col2a1-Cre^ mice [139]. The effectiveness of intra-articular injection of AAV-Runx1 has also been demonstrated in aging mice with surgical OA [140]. 

### 8.3. Runx3 Enhances Chondrocyte Maturation but Protects OA Development

Runx3 is expressed in prehypertrophic chondrocytes in the growth plate [20,141,142], and Runx3 expression is upregulated during chondrocyte differentiation and reduces Col2a1 expression and upregulates Col10a1 expression in vitro [89,142]. Furthermore, Runx3 has a redundant function with Runx2 and is involved in chondrocyte maturation [20]. These findings indicated that Runx3 enhances chondrocyte maturation. However, Runx3 regulates Acan expression [143]. Runx3 is expressed in all layers of the articular cartilage, and its expression is reduced after surgical OA. The deletion of Runx3 using Col2a1-CreERT2 or Prg4-CreERT2 accelerated OA development in surgical OA, whereas OA development in Runx3^fl/fl;Col2a1-CreERT2^ mice was not observed in an aging model of OA. Prg4 expression was reduced in the superficial zone of articular cartilage, while Acan expression was reduced in the deep zone in Runx3^fl/fl;Col2a1-Cre^ mice in surgical OA, and the adenoviral induction of Runx3 reversed these reductions. Runx3 appears to regulate the expression of Prg4 and Acan directly [126]. Thus, Runx3 functions in the growth plate and articular cartilage are different. 

### 8.4. Cbfb Protects OA Development

Kartogenin, a small molecule that promotes chondrocyte differentiation, has been shown to alleviate surgical OA. Kartogenin binds to filamin A, disrupts Cbfb-filamin A binding, and increases Cbfb-Runx1 binding [144]. Cbfb expression was reduced in patients with OA, and the articular chondrocyte-specific deletion of Cbfb using Gdf5 (growth differentiation factor 5)-Cre resulted in age-dependent spontaneous OA and acceleration of surgical OA. Col2a1 and Acan expression was reduced, whereas Mmp13 expression was increased in Cbfb^fl/fl;Gdf5-Cre^ mice. Tgfβ1 increased Col2a1 expression in articular chondrocytes from control mice but not in Cbfb^fl/fl;Gdf5-Cre^ mice. Cbfb1 protects Runx1 from polyubiquitination-mediated proteasomal degradation, and the Runx1/Cbfb complex is required for Col2a1 induction by Tgfβ signaling [145]. Furthermore, Cbfb^fl/fl;Col2a1-CreERT^ mice developed spontaneous OA and accelerated surgical OA. In the articular cartilage of Cbfb^fl/fl;Col2a1-CreERT^ mice, Tgfβ and Hippo/Yap signaling was reduced and Wnt signaling was increased, and the intra-articular injection of AAV-Cbfb increased Yap protein, reduced active Ctnnb1 (β-catenin) and protected OA progression, indicating that Cbfb protects OA through decreasing the Wnt/β-catenin signaling pathway and increasing the Hippo/Yap and Tgfβ/Smad2/3 signaling pathways [146]. However, Cbfb expression has been shown to be increased in OA patients. Furthermore, the overexpression of Cbfb reduced chondrocyte proliferation and increased chondrocyte apoptosis and Mmp13 expression, while the silencing of Cbfb showed the opposite results [147]. Although controversial, Cbfb appears to protect against OA development. Although Cbfb is required for the DNA binding capacity and protein stability of all Runx family transcription factors, its dependency on Cbfb differs among Runx family transcription factors in various skeletal tissues [91]. This may explain the protective effects of Cbfb on OA progression. 

## 9. Conclusions and Perspectives

Runx2 and Sp7 are needed not only for osteoblast and chondrocyte differentiation but also for the functions of differentiated osteoblasts. Runx2 is required for the expression of major bone matrix protein genes, and Sp7 regulates the formation of osteoblast/osteocyte processes. Sp7 is responsible for rare cases of OI, and the Runx family transcription factors and their cofactor Cbfb play important roles in the acceleration or protection of OA progression. Although ChIP, ATAC, and mRNA sequences of tissue-specific Runx2- or Sp7-deficient mice and their control mice revealed many Runx2 and Sp7 target gene candidates, their physiological significance needs to be elucidated individually. The appropriate upregulation or activation of Runx2 or Sp7 increases bone mass but accelerates OA progression. Therefore, targeting Runx2 and Sp7 specifically to the bone is important for the development of osteoporosis drugs, while inhibiting Runx2 specifically in articular cartilage is important for the development of OA drugs. Thus, the elucidation of the regulatory mechanisms of Runx2 and Sp7 gene expression in osteoblasts and chondrocytes by enhancers is important for the drug development. The development of a drug delivery system targeting the bone or articular cartilage is also necessary. 

## Figures and Tables

**Figure 1 ijms-25-10102-f001:**
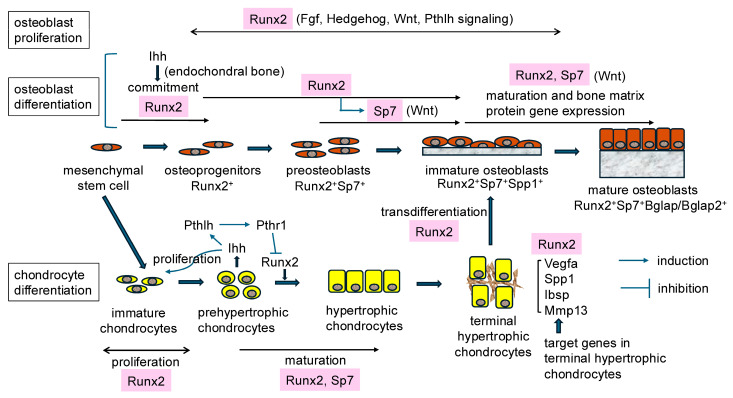
Functions of Runx2 and Sp7 in the proliferation and differentiation of osteoblast-lineage cells and chondrocytes: Runx2 induces the proliferation of osteoblast-lineage cells and chondrocytes and the commitment of mesenchymal stem cells to osteoprogenitors and their differentiation. Ihh is required for Runx2 expression in perichondrial mesenchymal cells during endochondral ossification. Runx2 induces the maturation of chondrocytes and the expression of Vegfa, Spp1, Ibsp, and Mmp13 in terminal hypertrophic chondrocytes and is required for the transdifferentiation of terminal hypertrophic chondrocytes into osteoblasts. Runx2 induces the expression of Ihh, which enhances chondrocyte proliferation, in prehypertrophic chondrocytes, and Ihh induces the expression of Pthlh, which inhibits Runx2 expression through Pthr1, forming a negative feedback loop. Sp7 induces the differentiation of preosteoblasts into osteoblasts and chondrocyte maturation.

**Figure 2 ijms-25-10102-f002:**
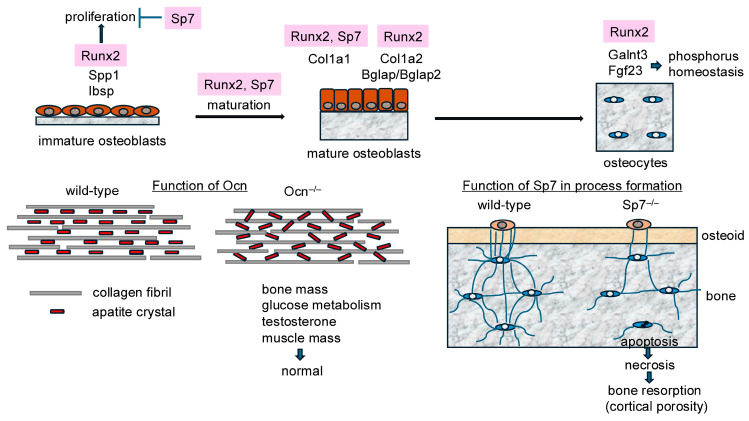
Functions of Runx2 and Sp7 in differentiated osteoblasts and osteocytes: Runx2 enhances the proliferation of immature osteoblasts, whereas Sp7 inhibits it, and both are required for osteoblast maturation. Runx2 induces the expression of Spp1 and Ibsp in immature osteoblasts, while Runx2 and Sp7 induce the expression of Col1a1. Runx2 also induces the expression of Bglap/Bglap2 (osteocalcin; Ocn) in mature osteoblasts, and Galnt*3* and Fgf23, the proteins that regulate phosphorus homeostasis, in osteocytes. Ocn is required for the alignment of apatite crystals parallel to collagen fibrils. Ocn does not regulate bone mass, glucose metabolism, testosterone synthesis, or muscle mass. Sp7 regulates the formation of osteoblast/osteocyte processes, and a reduction in the number of osteocyte processes causes osteocyte apoptosis and secondary necrosis, leading to cortical porosity.

**Figure 3 ijms-25-10102-f003:**
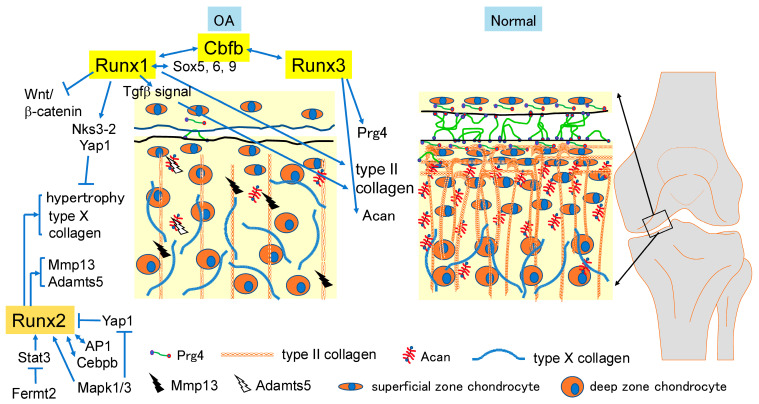
Pathogenic and protective features of Runx family transcription factors in OA: Chondrocyte maturation (hypertrophy) is a major cause of OA. Chondrocyte maturation leads to an increase in type X collagen and the expression of enzymes such as Mmp13 and Adamts5, which digest type II collagen and Acan, respectively, and a decrease in Prg4, type II collagen, and Acan. Runx2 induces chondrocyte maturation and the expression of Mmp13 and Adamts5. Yap1 (yes-associated protein 1) inhibits Runx2 and Fermt2 (fermitin family member 2, Kindlin-2) inhibits Stat3, which induces Runx2 expression. The activation of Mapk1/3 (mitogen-activated protein kinase 1/3) activates Runx2 and inhibits Yap1. Runx2 interacts with AP1 or Cebpb and induces Mmp13 expression. Runx1 increases Col2a1 expression through protein–protein interactions with Sox5, Sox6, and Sox9, and by activating Tgfβ signaling. Runx1 also inhibits chondrocyte maturation by inducing the expression of Nkx3-2 (NK3 homeobox 2) and increasing Yap protein levels, and it inhibits Wnt/β-catenin signaling, the excessive activation of which is associated with progressive joint damage [122]. Runx3 directly induces Prg4 and Acan expression. Cbfb stabilizes Runx1 and Runx3 proteins and protects against OA progression.

## Data Availability

No available data.

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
