# Peer review of "Regulation of Skeletal Development and Maintenance by Runx2 and Sp7"

_ijms, 2024, doi:10.3390/ijms251810102_

Round 1

Reviewer 1 Report

Comments and Suggestions for Authors

The manuscript provides a comprehensive overview of the roles of Runx2 and Sp7 in bone development, highlighting their interactions and regulatory mechanisms. The topic is timely and relevant, given the increasing interest in understanding the molecular basis of bone formation and its implications for bone-related diseases. Overall, the manuscript is well-structured and presents a clear narrative. However, there are several areas that require clarification and improvement.

ABSTRACT:

-        Abbreviations of genes/transcription factors must be accompanied by their full descriptions when first mentioned.

-        Abbreviations of genes/transcription factors should not be written in italics.

INTRODUCTION and MAIN TEXT:

-        The introduction effectively sets the stage for the discussion on Runx2 and Sp7. However, it would benefit from a more detailed explanation of the significance of these transcription factors in the context of bone pathologies, such as osteoporosis or osteogenesis imperfecta.

-        Abbreviations of genes/transcription factors must be accompanied by their full descriptions when first mentioned and I suggest to uniform them with the use of block letters throughout the manuscript.

-        While the manuscript cites several key studies, including more recent literature to provide a broader context for the findings and not only related to mice studies would be helpful. Specifically, studies from the last 2-3 years that explore novel regulatory mechanisms or interactions involving Runx2 and Sp7 should be incorporated.

-        Abbreviations of genes/transcription factors should not be written in italics.

-        Lines 71, 98, 115, 137, 157, 163: The content of these sentences is not reported in Figure 1.

-        Uniform the writing of the term osteocalcin throughout the manuscript and use its abbreviation consistently.

-        Line 234: clarify this process in Figure 2.

METHODOLOGY:

-        If applicable, please clarify the methodology used to gather the information presented in the review. For instance, were specific databases searched, and what criteria were used for selecting the studies included in the review?

FIGURES AND DIAGRAMS:

-        Abbreviations of genes/transcription factors should not be written in italics. Review this in the Figures’ captions and Figures themselves throughout the manuscript.

-        Figures should be inserted immediately after their first mention in the main text.

-        Figures 1 and 2 could be improved by inserting letters related to the explanations of the various steps, by improving the disposition of the upper arrows for better readability (Figure 1), and by contextualizing the bottom pictures (Figure 2).

-        The inclusion of improved or additional Figures or diagrams illustrating the signaling pathways and interactions between Runx2, Sp7, and other factors would greatly enhance the reader's understanding.

DISCUSSION:

-        The discussion section could be expanded to include potential therapeutic implications of targeting Runx2 and Sp7 in bone diseases. Discussing how modulation of these factors could lead to novel treatment strategies would add significant value to the manuscript.

CONCLUSION:

-        The conclusion summarizes the main points well, but it could be strengthened by emphasizing the future directions for research in this area. Suggestions for experimental approaches or unanswered questions in the field would provide a more forward-looking perspective.

REFERENCES:

-        Ensure that all references are up-to-date. Some references appear to be outdated, and including more recent studies would enhance the manuscript's relevance.

-        26 self-references out of 79 total references seem too many. Please consider retaining only relevant ones and include other authors’ studies.

LANGUAGE AND CLARITY:

The manuscript is generally well-written, but there are a few instances of jargon that may not be accessible to all readers. Consider simplifying complex terms or providing brief explanations to improve clarity.

Comments on the Quality of English Language

The manuscript could be simplified to enhance readability. Consider breaking down complex sentences into shorter, more straightforward ones. 

The use of scientific terminology is appropriate for the target audience. However, some terms may be too technical for readers who are not specialists in the field. It would be beneficial to provide brief definitions or explanations for less common terms to ensure accessibility.

Ensure consistency in the use of abbreviations and terminology throughout the manuscript. 

Some sections could benefit from smoother transitions between paragraphs and ideas. Adding transitional phrases can help guide the reader through the manuscript and improve the overall flow.

Author Response

Reviewer 1

Thank you for your useful and constructive suggestion.

The manuscript provides a comprehensive overview of the roles of Runx2 and Sp7 in bone development, highlighting their interactions and regulatory mechanisms. The topic is timely and relevant, given the increasing interest in understanding the molecular basis of bone formation and its implications for bone-related diseases. Overall, the manuscript is well-structured and presents a clear narrative. However, there are several areas that require clarification and improvement.

ABSTRACT: 

-        Abbreviations of genes/transcription factors must be accompanied by their full descriptions when first mentioned.

Full descriptions of gene names have been added.

-        Abbreviations of genes/transcription factors should not be written in italics. 

The gene names have been changed to roman-type.

INTRODUCTION and MAIN TEXT:

-        The introduction effectively sets the stage for the discussion on Runx2 and Sp7. However, it would benefit from a more detailed explanation of the significance of these transcription factors in the context of bone pathologies, such as osteoporosis or osteogenesis imperfecta.

The regulation of Runx2 and Sp7 expression has been added to the Introduction. The pathologies of skeletal diseases associated with RUNX2 and SP7 are described in Section 7.

-        Abbreviations of genes/transcription factors must be accompanied by their full descriptions when first mentioned and I suggest to uniform them with the use of block letters throughout the manuscript.

Full descriptions of abbreviations have been added, and gene names are described in Roman font.

-        While the manuscript cites several key studies, including more recent literature to provide a broader context for the findings and not only related to mice studies would be helpful. Specifically, studies from the last 2-3 years that explore novel regulatory mechanisms or interactions involving Runx2 and Sp7 should be incorporated.

I have added a section on osteoarthritis (Section 8). The regulatory mechanisms of Runx2 functions and interactions with Runx2 in OA development have been described in detail in recent studies.

-        Abbreviations of genes/transcription factors should not be written in italics.

They were written in Roman style.

-        Lines 71, 98, 115, 137, 157, 163: The content of these sentences is not reported in Figure 1.

This information has been added to Fig. 1 as much as possible.

-        Uniform the writing of the term osteocalcin throughout the manuscript and use its abbreviation consistently.

Osteocalcin is encoded by three genes, Bglap, Bglap2, and Bglap3. Bglap and Bglap2 are expressed in osteoblasts. We examined the expression of Bglap/Bglap2 as a marker for osteoblast maturation. As osteocalcin is usually abbreviated as Ocn, we used Ocn in section 5.3 to make it more understandable.

-        Line 234: clarify this process in Figure 2.

The functions of Runx2 and Sp7 in the proliferation of immature osteoblasts were added in Fig. 2.

METHODOLOGY:

-        If applicable, please clarify the methodology used to gather the information presented in the review. For instance, were specific databases searched, and what criteria were used for selecting the studies included in the review? 

Information was collected using PubMed, and papers including in vivo data were preferentially selected.

FIGURES AND DIAGRAMS:

-        Abbreviations of genes/transcription factors should not be written in italics. Review this in the Figures’ captions and Figures themselves throughout the manuscript. 

-        Figures should be inserted immediately after their first mention in the main text.

-        Figures 1 and 2 could be improved by inserting letters related to the explanations of the various steps, by improving the disposition of the upper arrows for better readability (Figure 1), and by contextualizing the bottom pictures (Figure 2).

Gene names are written in Roman font.

The locations of the figures have been corrected, and Fig. 1 and 2 have been improved.

-        The inclusion of improved or additional Figures or diagrams illustrating the signaling pathways and interactions between Runx2, Sp7, and other factors would greatly enhance the reader's understanding. 

I have added Fig. 3, which shows the pathogenic functions of Runx2 and the protective functions of Runx1 and Runx3 in OA progression.

DISCUSSION:

-        The discussion section could be expanded to include potential therapeutic implications of targeting Runx2 and Sp7 in bone diseases. Discussing how modulation of these factors could lead to novel treatment strategies would add significant value to the manuscript.

The cleidocranial dysplasia caused by RUNX2 mutations and osteogenesis imperfecta caused by SP7 mutations are described in section 7, and the progressive or protective functions of Runx family transcription factors and Cbfb in osteoarthritis are described in section 8. In Conclusions and Perspectives, treatment strategies using Runx2 and Sp7 for osteoporosis and osteoarthritis have been described.

CONCLUSION:

-        The conclusion summarizes the main points well, but it could be strengthened by emphasizing the future directions for research in this area. Suggestions for experimental approaches or unanswered questions in the field would provide a more forward-looking perspective.

I have added future directions for drug discovery for osteoporosis and osteoarthritis using Runx2 and Sp7 to the Conclusions and Perspectives.

REFERENCES:

-        Ensure that all references are up-to-date. Some references appear to be outdated, and including more recent studies would enhance the manuscript's relevance.

Recent studies have been added to the references section.

-        26 self-references out of 79 total references seem too many. Please consider retaining only relevant ones and include other authors’ studies.

I have reduced the number of self-citations and have added many references by other authors.

LANGUAGE AND CLARITY:

The manuscript is generally well-written, but there are a few instances of jargon that may not be accessible to all readers. Consider simplifying complex terms or providing brief explanations to improve clarity.

Brief explanations have been added to the revised manuscript.

Comments on the Quality of English Language

The manuscript could be simplified to enhance readability. Consider breaking down complex sentences into shorter, more straightforward ones. 

The use of scientific terminology is appropriate for the target audience. However, some terms may be too technical for readers who are not specialists in the field. It would be beneficial to provide brief definitions or explanations for less common terms to ensure accessibility.

Brief definitions and explanations have been added as much as possible.

Ensure consistency in the use of abbreviations and terminology throughout the manuscript. 

I have used the same abbreviation and terminology throughout the manuscript, except for Bglap/Bglap2. Osteocalcin (Ocn) was used in the section on the functions of Ocn because it is generally used for a long time and is more understandable to readers. 

Some sections could benefit from smoother transitions between paragraphs and ideas. Adding transitional phrases can help guide the reader through the manuscript and improve the overall flow.

I have tried to smoothen the transitions between paragraphs and ideas.

Reviewer 2 Report

Comments and Suggestions for Authors

In the past decade, numerous research papers have been published examining different aspects of Runx2 and Sp7 in bone development. However, these current papers often lack focus and a critical analysis of the underlying issues. Some specific comments:

1. The current Figure 1 contains only broad, textbook-level information on bone biology that is not sufficiently relevant for this review paper. Instead, the introductory figure should be more focused on the specific problems and topics that will be addressed throughout the review.

2. Figure 2 attempts to cover the diverse molecular mechanisms underlying osteocyte apoptosis and secondary necrosis, but this broad scope makes the information unfocused and uninformative. The figure contains only vague, general statements about factors involved in pathogenesis, without providing meaningful details. The Reviewer would recommend focusing the discussion on a few specific diseases and exploring those in greater depth, rather than trying to address the full range of mechanisms in a superficial manner.

3. The manuscript would benefit from a more direct exploration of the potential applications of Runx2 and Sp7 in disease diagnostics and treatment. This focus would likely be more relevant and engaging for the reader.

4. When discussing the research and clinical applications of Runx2 and Sp7, conclusions and future perspectives should take a more critical approach. Analyses should not only highlight the positive aspects, but also address the more problematic elements, especially regarding clinical use.

Comments on the Quality of English Language

1. The authors should also heed the advice to seek assistance from a professional with strong English language proficiency to ensure the overall quality and clarity of the writing throughout the paper 

Author Response

Reviewer 2

Thank you for your useful and constructive suggestion.

In the past decade, numerous research papers have been published examining different aspects of Runx2 and Sp7 in bone development. However, these current papers often lack focus and a critical analysis of the underlying issues. Some specific comments:

  1. The current Figure 1 contains only broad, textbook-level information on bone biology that is not sufficiently relevant for this review paper. Instead, the introductory figure should be more focused on the specific problems and topics that will be addressed throughout the review.

I have improved Figure 1 by adding more detailed information.

  1. Figure 2 attempts to cover the diverse molecular mechanisms underlying osteocyte apoptosis and secondary necrosis, but this broad scope makes the information unfocused and uninformative. The figure contains only vague, general statements about factors involved in pathogenesis, without providing meaningful details. The Reviewer would recommend focusing the discussion on a few specific diseases and exploring those in greater depth, rather than trying to address the full range of mechanisms in a superficial manner.

The pathogenesis of human skeletal diseases with RUNX2 and SP7 mutations is described in Section 7, and the progression or protection of osteoarthritis (OA) by Runx family transcription factors and Cbfb is described in Section 8. I have added Figure 3, which focuses on OA. The title was also changed to “Regulation of skeletal development and maintenance by Runx2 and Sp7.”

  1. The manuscript would benefit from a more direct exploration of the potential applications of Runx2 and Sp7 in disease diagnostics and treatment. This focus would likely be more relevant and engaging for the reader.

In the Conclusions and Perspectives section, I describe the future directions for the application of Runx2 and Sp7 in the development of drugs for osteoporosis and osteoarthritis.

  1. When discussing the research and clinical applications of Runx2 and Sp7, conclusions and future perspectives should take a more critical approach. Analyses should not only highlight the positive aspects, but also address the more problematic elements, especially regarding clinical use.

As described above, a problematic element for drug development targeting Runx2 and Sp7 is described in the Conclusions and Perspectives section.

Comments on the Quality of English Language

  1. The authors should also heed the advice to seek assistance from a professional with strong English language proficiency to ensure the overall quality and clarity of the writing throughout the paper.

The manuscript has been edited by the English Medical Service in Japan.

Round 2

Reviewer 2 Report

Comments and Suggestions for Authors

In the revised article, the authors comprehensively addressed the reviewer's comments by modifying the manuscript and figures, and providing detailed point-by-point responses.